# Advances in Nanotechnology-Based Biosensing of Immunoregulatory Cytokines

**DOI:** 10.3390/bios11100364

**Published:** 2021-09-30

**Authors:** Warangkana Lohcharoenkal, Zareen Abbas, Yon Rojanasakul

**Affiliations:** 1Essity Hygiene and Health AB, Mölndals Bro 2, SE-405 03 Gothenburg, Sweden; warangkana.lohcharoenkal@essity.com; 2Department of Chemistry and Molecular Biology, University of Gothenburg, Kemigården 4, SE-412 96 Gothenburg, Sweden; 3Department of Pharmaceutical Sciences, West Virginia University, Morgantown, WV 26505, USA; 4West Virginia University Cancer Institute, West Virginia University, Morgantown, WV 26505, USA

**Keywords:** cytokines, biosensor, nanomaterials

## Abstract

Cytokines are a large group of small proteins secreted by immune and non-immune cells in response to external stimuli. Much attention has been given to the application of cytokines’ detection in early disease diagnosis/monitoring and therapeutic response assessment. To date, a wide range of assays are available for cytokines detection. However, in specific applications, multiplexed or continuous measurements of cytokines with wearable biosensing devices are highly desirable. For such efforts, various nanomaterials have been extensively investigated due to their extraordinary properties, such as high surface area and controllable particle size and shape, which leads to their tunable optical emission, electrical, and magnetic properties. Different types of nanomaterials such as noble metal, metal oxide, and carbon nanoparticles have been explored for various biosensing applications. Advances in nanomaterial synthesis and device development have led to significant progress in pushing the limit of cytokine detection. This article reviews currently used methods for cytokines detection and new nanotechnology-based biosensors for ultrasensitive cytokine detection.

## 1. Introduction

The term ‘cytokine’ defines a large group of signaling proteins that exert diverse biological functions critical to normal homeostasis and disease development including innate and acquired immunity, hematopoiesis, inflammation, and proliferation [1,2]. Cytokines are secreted by many cell types and predominantly function in a paracrine manner by having effects on adjacent cells. They may also act at a distance via the circulation (endocrine or systemic effect) and on the cell of origin (autocrine effect). Cytokines function by binding to specific receptors on target cells, thereby activating intracellular signaling cascades that result in gene transcription and other cellular processes. Cytokines can be broadly classified based on their functions into major classes which include pro-inflammatory cytokines (e.g., interleukin-1 alpha; IL-1α, IL-6, tumor necrosis factor alpha; and TNF-α), anti-inflammatory cytokines (e.g., IL-1 receptor antagonist; IL1ra, and IL-10), neutrophil recruitment and activation cytokines (e.g., IL-1α, IL-8, IL17A, and TNF-α), eosinophil recruitment and activation cytokines (e.g., IL-2, IL-3, IL-4, and IL-5), cytokines derived from T-cells (e.g., interferon gamma; IFN-γ, IL-2, IL-6, IL-9, IL-12, IL-19, and IL-25), T-cell recruitment cytokines (e.g., IL-10), and growth factors (e.g., transforming growth factor-β and TGF-β) [3]. They can also be classified into different families based on the structure as given in Table 1. The effect of a particular cytokine is often a result of more than one function, and not restricted to only one biological function that is implied in its name [4]. 

Although cytokines are studied currently in nearly every field in biology, cytokine-mediated effects are predominantly studied in the field of inflammation. During the past few decades, cytokines have been extensively explored as diagnostic, prognostic and therapeutic targets in many inflammatory diseases [1]. Cytokines are either undetectable or present at very low levels in body fluids and tissues under normal circumstances. Therefore, their elevated expression levels indicate the activation of some specific pathways associated with disease processes (e.g., inflammation). Furthermore, the assessment of cytokines in body fluids, cells, and tissues can provide important information on the disease progression and development of treatment strategies. To date, a wide range of cytokine assays are available for the assessment of biological functions and therapeutic responses [6]. Biosensors have increasingly become a preferred alternative to standard detection methods due to many advantages such as high selectivity and sensitivity, quick turnaround time, cost effectiveness, ease of fabrication, user friendliness, adaptable nature, and ability to miniaturize [7]. Much attention has been given to the development of cytokine biosensors for point-of-care (POC) to facilitate rapid diagnosis and disease management. Ideal POC features, according to World Health Organization, are affordability, sensitivity, specificity, user-friendliness, rapidity, equipment-free, and deliverable to end users [8]. With the challenges associated with cytokine detection such as complex cytokine signaling, dynamic secretion, low concentration, and low stability, there has been a continuous development of detection platforms as well as materials for cytokine biosensors fabrication. A number of review articles on the cytokine biosensing topic have been published during the past decade. Most of the reviews focused on biosensors for specific cytokines [9,10] or diseases [11,12,13,14]. Some discussed the detection approaches, e.g., a review on cytokine immunosensors by Liu et al. [15] and an article about the current situation on electrochemical biosensors for cytokine profiling by Dutta et al. [16]. With recent advances in nanoscience and technology, many emerging technologies with superior sensing capabilities have been developed. Nanomaterials have been used in these technologies based on their unique optical, electrical, and mechanical properties related to their reduced dimension. Novel sensing platforms based on nanomaterials have shown considerably improved performances in cytokines analysis [17]. The application of nanomaterials for cytokine detection has been extensively reviewed recently by Singh et al. [17]. Several nanomaterials, including metal nanoparticles, zinc oxide nanorod, photonic crystal, and semiconducting nanoparticles have been discussed. However, only the optical mode of detection was taken into consideration. This review article discusses various cytokine detection techniques, as well as recent advances in cytokine detection strategies exploiting the unique properties of nanomaterials. Nanomaterials included in this review are quantum dots, noble metal nanomaterials, metal oxide nanomaterials, carbon-based nanomaterials, polymer nanomaterials, and bionanomatrials. For each highlighted nanomaterial, its detection capabilities, advantages, and challenges are discussed. 

## 2. Conventional Cytokines Measurements

Cytokines can be measured by a range of techniques, including the detection of soluble cytokines (e.g., ELISA), cytokines produced by a population of cells or single cell (e.g., flow cytometry, enzyme-linked immunospot (ELISPOT), intracytoplasmic cytokine staining (ICC), and mRNA-based assays), cytokines in tissues (e.g., immunostaining), and the simultaneous detection of multiple cytokines in multiplex manner (e.g., DNA and protein microarrays) [6]. Advantages and disadvantages of the listed techniques are summarized in Table 2.

### 2.1. Enzyme Linked Immunosorbent Assay (ELISA)

Immunoassay using enzyme-conjugated antibodies has been recognized as a gold standard for cytokine detection and there are numerous commercially available ELISA kits for various cytokines. The basic sandwich-type ELISA employs highly purified antibodies (capture antibodies) which are coated on microwell plates. The cytokine presented in the samples is specifically captured by the immobilized antibodies and detected by enzyme-conjugated antibodies (detection antibodies). A colored product is generated by the enzyme after the addition of a suitable chromogenic substrate. The intensity of the colored product correlates with the level of the cytokine being detected, which can be quantified by spectrophotometry. The most important factor for the success of this method is the quality of the antibodies. Some cytokines exist in multiple forms (e.g., monomeric or polymeric, precursor or degradation products) which bind differently in immunoassays. Non-specific binding of capture antibodies with other proteins in the sample is also an important factor influencing the accuracy of cytokine measurements [18].

### 2.2. Radioimmunoassay (RIA)

RIA is a method for cytokines detection in biological samples. This method requires a sample with an antigen of interest, a complementary antibody, and the radiolabeled version of antigen (usually with ^125^I label). In this assay, the sample and antibody are incubated together before the radiolabeled antigen is added to compete and displace the sample antigen. The antigen–antibody complex is then pelleted and radioactivity of the pellet can be quickly and easily counted on a beta scintillation counter (for plate format assay) or a gamma counter (for tube format assay) [19]. This assay does not use enzymes, resulting in a lower risk of interference from the sample [6].

### 2.3. Enzyme-Linked Immunospot (ELISPOT)

ELISPOT is currently the method of choice for the detection of cytokines produced by a single cell. It is commonly performed in a well plates format in the same way as ELISA, which is adaptable to high throughput assays. The results (e.g., number of spots per well) can be acquired by unbiased image analysis [20]. An increasing number of modified ELISPOT assays has been developed, e.g., the dual-color ELISPOT assay for the differentiation of three subtypes of cells based on their secretion of two different cytokines. However, this assay has some limitations related to the difficulties in interpreting mixed color spots from the cell that secretes both cytokines, so there has been an attempt to develop the Fluorospot assay using multiple antibodies [21]. With a specialized reader that employs a light microscope and fluorescence illumination, this assay provides an improved discrimination of the dual cytokine-producing cells [22]. 

### 2.4. mRNA Based Assays

Quantitative real-time PCR (qRT-PCR) is a powerful technique for quantitative assessment of a particular mRNA transcript relative to a housekeeping transcript [23]. To date, cytokine-specific mRNA assays using biotin-labeled capture probes and digoxigenin-conjugated detection probes in a microplate format are available commercially [24]. The main disadvantage of mRNA-based assays is the requirement for high quality mRNA, which is highly susceptible to degradation by endogenous RNases. It is also worth taking into consideration that the absence of mRNA could reflect their rapid processing and/or degradation than the lack of synthesis [20]. 

### 2.5. Immunostaining

This technique can be used to detect different cytokines in cells and tissues. It can be used for in situ detection of mRNA or proteins using suitable probes. Similar to other immunoassays, this assay is very much dependent on the selected antibodies. As the technique includes a fixation step, it might alter cytokine conformation or cause partial denaturation that makes the cytokine unrecognizable by the antibodies [25]. There is also a possibility to observe nonspecific staining which can be controlled by using isotype-matched control antibodies. Immunostaining can be detected either by fluorescence or light microscopy using fluorescent or enzyme labeled antibodies. 

### 2.6. Intra-Cytoplasmic Cytokine Staining (ICC) 

Cytokine detection by ICC can be performed by flow cytometry, typically by using detergent-permeabilized cells to allow intracellular entry of antibody probes. Both fixation and permeabilization steps can lead to some artifacts which can be avoided by carefully titrating the antibodies and using appropriate control antibodies. Data should be interpreted with caution, as the assay cannot distinguish the cells that produce cytokines from those that merely internalize them [26]. ICC has been widely used in determining cytokine-producing T cells in circulation or stimulated hematopoietic cells by simultaneously detecting cell surface markers and specific cytokines [27]. 

### 2.7. Cytokine Microarrays

Due to the complexity of cytokine signaling pathways, much attention has been given to the development of assays that can simultaneously detect multiple cytokines in a sample. Based on the ELISA principle, multiplex cytokines microarrays have been developed and are now commercially available by several vendors. Expression profiling of multiple cytokines by microarrays has a tremendous amount of applications as it can generate a vast amount of information that can aid the understanding of the disease process. However, reproducibility of the multiplex assays could be problematic as many commercial assays show poor precision, high variation, and impaired sensitivity of detection of the individual analytes. The involvement of a large number of analytes in miniaturized microarray formats makes it prone to cross reactivity and false positive results [28]. 

In addition to protein microarrays, DNA microarrays have been used to determine changes in cytokines’ gene expression in response to various stimuli and pathologic conditions [29]. This technique involves spotting thousands of nucleic acid probes at known locations on a slide called a gene chip. Complementary pairing between the sample’s DNA and specific nucleic acid sequences on the chip produces corresponding measurable signals (e.g., fluorescence or chemiluminescence) [30]. Although this method has been used to validate the involvement of cytokines in various inflammatory diseases and there has been an increase in the availability of DNA microarrays commercially, it is still largely restricted to research practices. 

## 3. Biosensor (lab-on-a-Chip) Concept for Cytokines Detection 

Cytokines are widely used for the monitoring and prediction of disease progression and therapeutic response. Currently, ELISA is the most routinely used cytokine assay for clinical samples analysis. Although relatively inexpensive and amenable to multiplexing, ELISA-based assays are not suitable for POC usage as it typically takes hours to prepare, process, and read the samples. It also requires a large sample size and relies on bulky equipment that is not suitable for POC applications [16,31]. With the limitations of conventional assays and the need for critical diagnostic cytokine measurements, many biosensor-based technologies have been developed as a simple, sensitive and inexpensive tool for cytokine detection. In particular, the development of portable handheld or wearable devices that can monitor health status continuously has received the most attention. 

Biosensors are analytical devices that can convert specific biorecognition events into quantifiable signals. They are generally composed of the receptor (e.g., enzyme, antibody, or DNA) which specifically recognizes the analyte or biomarker (e.g., enzyme substrate, antigen, or DNA) and the transducer which converts the binding interaction into a measurable signal proportional to the analyte’s concentration (Figure 1). Biosensors are generally classified based on their signal detection mechanisms, e.g., electrochemical, optical, piezoelectric, or magnetic. These technologies have broad applications in health, environmental and forensic sciences [14]. In recent years, the emerging role of inflammation in many disease processes has led to the development of advanced biosensors for specific inflammatory cytokine detection in clinical samples (e.g., body fluids or tissue samples from patients). Key criteria for the development of these biosensors include sensitivity, selectivity, dynamic range, analysis time, accuracy, precision, and lifetime. Sensitivity and selectivity are the two most important factors for biosensor applications in research laboratory settings. Beyond the bench-top application, other factors, e.g., multiplexing, user friendliness, cost, and real-time monitoring become important. The configurations and detection approaches with the possibility to be integrated to the existing technologies or devices in clinical and POC settings are also much sought after [17]. 

An example of recently developed cytokine biosensors is the electrochemical biosensor for IL-1β and IL-10 detection in patients with heart failure after surgical implantation of left ventricular assist devices (LVADs). The non-biocompatibility nature of LVADs resulted in acute inflammation caused by the release of these two cytokines. The developed biosensor can detect both cytokines at a low level (e.g., 1–15 pg/mL), allowing early detection of the inflammation [32]. Another example is the electrochemical biosensor for the detection of IL-12 in multiple sclerosis patients based on the evidence that its high level in cerebrospinal fluid and serum correlates well with disease progression. It is also tested for the early identification of relapse and prediction of therapy outcome, which is of clinical importance for therapeutic intervention [33]. For multiplex cytokine detection, there has been an effort to develop a microfluidic surface plasmon resonance (SPR) biosensor for the detection of multiple analytes based on refractometry technique. This biosensor can perform multiplex analysis of six cytokines (IL-2, IL-4, IL-6, IL-10, IFN-γ, and TNF-α) within a linear detection range of 5–20 pg/mL, and only requires 1 μL of serum sample. The device was used for real-time monitoring of these inflammatory cytokines in two neonates after the bypass surgery [13]. In another study, a thumbnail-size patch was developed to detect seven cytokines from skin surface using self-assembly microdisks containing antibody arrays. This array-based patch was shown to provide sensitive detection of five cytokines, namely IL-1α, IL1RA, IL-17A, TNF-α, and IFN-γ [34].

## 4. Nanomaterials and Biosensing Application 

As outlined in the previous section, several biosensors have been developed for the detection of various cytokines using different optical, electrical, electrochemical, and hybrid transduction methods. Among the various materials employed in these biosensors, nanomaterials have received the most attention due to their small size, ability to miniaturize, portability, and improved detection capabilities [17]. 

Nanomaterials are a group of materials that have nanoscale size (i.e., less than 100 nm in at least one dimension). They normally have different physicochemical properties than the same bulk materials. Nanomaterials can be classified into zero-dimensional (0D), 1D, and 2D. The shape of nanomaterials can vary from nanoparticles (NPs), nanorods (NRs), nanotubes (NTs), nanowires (NWs), and ultrathin films [35]. Nanomaterials are generally used as a transducer material, which is a key component of biosensors but can also be used for other functions such as immobilization support and signal amplifier [36]. Owing to the high reactive surface area and facile surface functionalization, improved performance of biosensors such as increased sensitivity of several orders of magnitude can be achieved. In some cases, the enhanced sensitivity is due to the fact that nanomaterials are of a similar size as the analyte, and thus are capable of probing previously unreachable matrices [37]. Biocompatibility is an important factor when designing a biosensor for biological applications [38]. Selectivity of the biosensors to specific analytes can be tailored by immobilizing biological recognition elements (e.g., enzymes, antibodies, and receptors) on the sensor substrate [39]. To date, many kinds of nanomaterials, such as metal NPs, metal oxide nanomaterials, and carbon NPs, have been used for biosensors fabrication. These nanomaterials can have different functions in different sensing systems [40]. 

Optical, electrochemical, and magnetic transduction methods are three major signal transduction methods in nanomaterial-based biosensors. The optical methods, particularly those that use colorimetric techniques to detect the signal in visible spectrum, are ideal for general users. Electrochemical methods are highly specific and can be simplistic to miniaturize [41]. Magnetic transduction methods exhibit minimal background signal as compared to optical and electrochemical methods, making them ideal for the detection of analytes at low concentration. In some applications, magnetic nanomaterials are used to pre-concentrate the analytes prior to their detection by optical or electrochemical methods.

### 4.1. Optical Transduction

Optical transduction is based on the emission or absorption of optical signal in a sample under irradiation (i.e., by visible, ultraviolet, or infrared light) [42]. To date, fluorescence and SPR-enabled spectroscopies are the two common optical transduction methods employed in the nanosensor field.

Fluorescence spectroscopy measures the emission of a fluorophore after excitation. In nanosensor applications, quantum dots (QDs), polymeric NP probes, or dye-doped silicon are frequently employed due to their superior photostability and robustness compared to traditional fluorescent dyes [43,44]. Changes in the fluorescence signal (i.e., quenching or restoration of the signal) can be interpreted as a result from the interaction between the analyte and NP or a conformational change of the sensor.

SPR-enabled spectroscopy is based on the localized SPR (LSPR) of noble metal nanomaterials [45]. The interaction of biomolecules with the nanomaterials results in the changes in LSPR and the optical response. Due to the sensitivity of LSPR band to the interparticle distance, it can be used to differentiate the dispersion and aggregation. It is commonly used in combination with secondary spectroscopic technique to generate a surface enhanced spectroscopy, i.e., surface-enhanced fluorescence (SEF) or surface-enhanced raman spectroscopy (SERS) [44].

### 4.2. Electrochemical Transduction

This method measures the change in electrical current or potential that results from the interaction between the analyte and the electrode. It can be classified into amperometric, potentiometric, and conductometric methods. Amperometric biosensors are based on the oxidation/reduction reaction that generates an electrochemical signal under a specific potential. The amperometric technique is generally used in the design of electrochemical biosensors by measuring current between the two electrodes when a potential is applied, or the analyte undergoes a redox reaction. For potentiometric biosensors, ion-selective electrodes are used to transduce biological reaction into electrical signal. The performance of this detection method is much dependent on the transducer surface properties, which determine the relative potential of the working electrode to reference electrode when zero current flows between them. This detection method is affected by the variations in ionic strength and pH of the detection media [46]. Conductometric biosensors measure the electrical current between the electrodes and reference. The current is generated due to the conductivity property of the analyte. It has mostly been used for enzyme detection to measure the change in ionic strength and conductivity of a solution between two electrodes as a result of enzymatic reactions [39]. Nanosensor designs can involve modifications of the electrodes (e.g., gold, silver, platinum, and graphite) with nanocarbons (e.g., carbon nanotubes and graphene) or functionalization with recognition elements (e.g., antibodies and aptamers) [41].

Due to the portability, self-containment, and low cost, electrochemical biosensors are dominant analytical tools for the biomarker detection. The application of functionalized nanomaterial for the fabrication of sensitive and selective electrochemical biosensors has been extensively explored in the biomedical field and other areas of analytical sciences. Numerous electrochemical biosensing techniques have been developed based on unique nanostructure designs for signal amplification and improved signal-to-noise ratios compared to the current electrochemical techniques [39,41,47]. The size and morphology of the analytes also affect sensor function. Small analytes are generally more diffusive and accessible, thereby better detection efficiency can be achieved compared to large molecules [48]. However, further knowledges on effective integration of nanomaterials, electrode microfabrication, and sensor engineering are crucial for the development of next-generation cytokine biosensors. 

### 4.3. Magnetic Transduction

Magnetic NPs (MNPs) has been explored for the detection of analytes in biological samples. MNPs generally have low background magnetic signal. Under an applied magnetic field, magnetic signal from MNPs can be collected regardless of the optical properties of the solution [44,49]. MNPs has also been used to concentrate, separate, and purify analytes prior to the application of the secondary transduction method, such as electrochemical stripping [49]. 

An example of magnetic transduction is the magnetic-relaxation switches that incorporate superparamagnetic iron oxide NPs with the principle that the individual nanomagnetic probes will cluster upon their interaction with the target. The obtained NP clusters can enhance dephasing of the spins of the surrounding water protons. The change in spin–spin relaxation can be detected by magnetic resonance relaxometry [50]. Thus far, magnetic relaxation switches have been used for the detection of various analytes, e.g., DNA, mRNA, proteins, and viruses [51,52].

## 5. Nanomaterials as Biosensor for Inflammatory Cytokines

### 5.1. Quantum Dots (QDs)

QDs are semiconductor nanocrystals in a size range of 2–10 nm with unique optical and electronic properties. They are commonly synthesized from chemical elements in group II–VI or III–V of the periodic table, such as cadmium selenium (CdSe) or cadmium telluride (CdTe) [53]. Due to the quantum confinement effect, QDs show very different optoelectronic properties from bulk materials and have been extensively used in various research areas such as bioimaging, drug monitoring, diagnostics, and biomolecular interactions [54,55]. QDs typically have broad absorption spectra covering the ultraviolet to visible wavelength, depending on their particle size. The broad excitation and narrow size-tunable emission spectra of QDs, as well as their negligible photobleaching and high photochemical stability, make them superior alternatives to traditional fluorophores in many applications, including biosensor development [55]. The wide absorption spectrum of QDs offers great advantages in tracing multiple analytes simultaneously using simple instruments. QDs also offer some advantages in electrochemical applications with their semiconductor properties [56]. 

Recent studies demonstrated the value of QDs as a sensitive sensor for cytokines. Xu et al. reported a very low detection limit of 3.36 fg/mL for IL-8 when using DNA-templated QDs (DNA-QDs) as an electrochemical probe. In this application, samples were treated with a reducing agent, tris-(2-carboxyethyl) phosphine, and the obtained active thiols were then captured by the antibody-functionalized magnetic beads. DNA-QDs were then modified with maleimide, which can react to the active thiol. The amount of IL-8 in the samples can be determined from the electrochemical response of DNA-QDs [57]. In another study, graphene quantum dots (GQDs) were used to detect ultra-small amounts of cytokine intracellularly. The GQDs can be modified with cytokine aptamers (Ap-GQDs) or epitopes (Ep-GQDs). Both modified GQDs fluoresce in solutions at sufficient dilution; however, their fluorescence is quenched when conjugates of the modified GDQs are formed, i.e., due to aggregation between Ap-GQDs and Ep-GQDs. In the presence of cytokine-secreting cells, the secreted cytokines can compete with the epitope and cause the disassociation of the Ap-GQDs and Ep-GQDs aggregates, and therefore recover fluorescence signal. The Ap-GQDs and Ep-GQDs conjugates were successfully used to detect intracellular IFN-γ level at concentrations as low as 2 pg/mL [58]. 

### 5.2. Noble Metal Nanomaterials (NMNs)

Noble metals are metallic elements that are highly resistant to corrosion and oxidation, such as gold, silver, palladium, ruthenium, platinum, rhodium, osmium, and iridium. Currently, there is an unprecedented expansion of research to explore the application of NMNs in biotechnology and biomedicine due to their attractive properties, including a large surface area that enhances their biorecognition and receptor immobilization capabilities, the ability to catalyze electron transfer reactions, and good stability in biological systems [59]. NMNs have been used in sensors to amplify signal and improve sensitivity of the detection of biomolecules [60]. Optical, electrochemical, and piezoelectric sensing methods are the main sensing methods for NMN-based sensors. Among these biosensing methods, the colorimetric method has been the most frequently investigated due to its simplicity and strong prospect for applications at POC. For this reason, most biosensors to date are based on gold and silver NPs as they possess optical properties in the range of visible wavelength. In addition, they are easy to synthesize and functionalize. Other NMNs that have been used in biosensing applications include platinum NPs, albeit to a much lesser extent, due to their unique electrochemical properties [61].

• Gold Nanoparticles (AuNPs)

The majority of NP-based biosensing applications are based on AuNPs due to their optical properties [62,63]. AuNPs can transfer electrons efficiently, and the electron transfer rate in the presence of AuNPs is about 5000 per second compared to 700 per second in their absence [64]. The strong light-scattering and electromagnetic-enhancing properties of AuNPs allow them to be used as signal amplifier in diverse biosensor applications [65]. One interesting optical property of gold surfaces is that irradiation at a specific wavelength can cause oscillation of the electrons in the conduction band called resonant surface plasmons. This phenomenon is strongly dependent on size and shape of the particles, as well as dielectric constant of the environment. The change in oscillation frequency due to the recognition events provides unique advantage in the detection based on the color change of AuNPs, which is observable with the naked eye [66]. 

Research on the application of AuNPs as cytokine biosensors during the past decade can be divided into stages from proof of concept, attempts to improve existing detection methods, and development of new technologies or devices. For the proof of concept, there have been many studies investigating the SPR property of AuNPs for improved detection of several cytokines, including IL-1, IL-6, IFN-γ, and TNF-α [67,68,69]. One study demonstrated the detection of IL-6 based on visible color change caused by AuNPs aggregation (Figure 2a) [70]. Several other studies reported the use of AuNPs to improve existing cytokine detection methods, including immuno-polymerase chain reaction (iPCR)-based detection of IL-3 [71]. In this system, a DNA template and a target antigen-specific antibody are conjugated through AuNPs to form a new detection reagent with superior sensitivity and detection range than standard ELISA assay. The AuNP-iPCR assay is also easier and faster to perform with the possibility to measure low concentration and complex samples. New development of AuNP-based biosensors has focused on device portability. One study employed nanoimprinted gold strips with colloidal AuNPs for IL-10 detection. The sensor was fabricated via nanoimprinting process on polyethylene terephthalate (PET) film. AuNPs were deposited on the PET film to obtain the structures with a strong LSPR extinction peak. IL-10 detection antibody was subsequently immobilized on the strip surface. Colloidal Au nanocube (AuNC) was used to enhance the LSPR signal and the assay with AuNC crosslinked with IL-10 antibody demonstrates the sensitivity of detection at the nanomolar level [72]. In another study, a cytokine detection device using AuNP-modified silica optical fiber was developed to monitor locally variable concentrations of cytokine IL-6. The fiber was introduced into an intrathecal catheter. There were a micrometer-sized holes along the fiber length for fluid exchange. The optical fiber was modified with AuNPs layer and functionalized with IL-6 capture antibody. IL-6 detection antibody loaded on fluorescently labeled magnetic NPs was used for the detection (Figure 2b). The device managed to detect as low as 1 pg/mL of IL-6 from low volume sample (1 µL). It can also measure secreted IL-6 in real time, which is desirable for health condition monitoring [73].

• Silver Nanoparticles (AgNPs) 

AgNPs have received enormous attention in the field of biosensors development due to their catalytic properties and enhanced detection of clinical markers [74,75]. Silver has many advantages compared to gold, such as sharper extinction bands, higher extinction coefficients and ratio of scattering to extinction, and extremely high field enhancements. However, it has been far less employed in sensors than gold due to its lower chemical stability in biological media and significant cytotoxicity compared to AuNPs [76,77]. Efforts have been made to improve the stability of AgNPs, i.e., by surface coating or modification. As a result, AgNPs are gaining popularity, and several research groups have begun to explore their use in optical sensors [76]. One of the most promising sensors developed from AgNPs is based on the enhancement of electromagnetic fields, which results in the so-called surface enhanced spectroscopies such as SERS and metal enhanced fluorescence (MEF). SERS allows ultrasensitive detection of various compounds, from chemical pollutants to biomolecules. It is sensitive enough to detect a single molecule of the analyte. In addition, this technique can give information about the structure of the analyte. Encapsulated SERS tags have been explored as alternative fluorescence tags. These SERS tags, compared to conventional fluorescence tag, demonstrate better stability and multiplexing capability. With this approach, Zhou et al. developed a 15-plexed silver-enhanced sandwich immunoassay (SENSIA). AgNPs linked to secondary antibodies are used, and the detection method was based on the quantification of the deposited Ag. This resulted in the affordable colorimetry-based detection for IL-1β, IL-10, IL-2, IL-4, and IL-6. The detection step can be performed using a simple flatbed scanner instead of the array readers that are normally used for multiplex cytokine detection with ELISA kits (Figure 3) [78]. Similar to SERS, MEF also has a great potential to overcome the drawbacks of traditional fluorescence tags [76]. In the vicinity of the fluorophores (4–10 nm distance), AgNPs can enhance the luminescence of fluorophores while other metals typically quench them. AgNPs, when deposited on a glass substrate, have shown to increase the emission intensity of a commonly used biological probe fluorescein by at least 3-fold [79]. Generally, MEF can lead to a 10–1000-fold intensity enhancement. The enhancement efficiency is based on the distance between AgNPs and the dye, the aggregation of AgNPs, and the overlap between the LSPR band of the AgNPs and the emission spectra of the fluorophore [76,80,81,82]. A study by Szmacinski et al. exemplified this approach for TNF-α detection. With a plasmonic nanostructure containing a unique composition of silver and silica layers, a more than 200-fold increase in fluorescence signal in the immunoassay was achieved. The system was used to image TNF-α secretion in primary murine macrophages after stimulation with lipopolysaccharide. The MEF assay also provided advanced imaging capability for the detection of cytokine secreted from a single cell without any complicated procedures required by standard methods [83].

• Palladium Nanoparticles (PdNPs)

PdNPs have captivated much attention in biomedical sensor development due to their exceptional catalytic activity. A variety of PdNPs with diverse composition and functionalization have been investigated for the detection of different biomarkers during the last decade. Higher abundance of Pd compared to other noble metals makes it a cheaper alternative for application in biosensing platforms [84]. Pd is a plasmonic material that possesses higher refractive index sensitivity than AuNPs and AgNPs. PdNPs are however not ideal in plasmonic sensing due to the broad extinction spectra and low scattering efficiency. They are therefore mostly investigated in combination with other NMNs for the optimization of sensor performance [85]. For example, PdNPs have been used with platinum (Pt) NPs to generate bimetallic Pd–Pt NPs for ultra-sensitive detection of IL-6 [86]. In this application, Pd–Pt NPs were synthesized and incorporated in an immunosensor to act as a functionalized bionanolabel. This immunosensor exhibited high sensitivity and selectivity for IL-6 detection with a broad linear response range of 0.1 to 2000 pg/mL and detection limit of 0.032 pg/mL in serum samples [86].

### 5.3. Metal Oxide Nanomaterials

Metals can form a large diversity of oxide compounds called metal oxides. Due to their exceptional electrical, magnetic, mechanical, optical, and catalytic properties, metal oxide-based nanomaterials and their polymeric composites have been used for many applications, including biosensing. Varieties of metal oxide NPs, including titanium dioxide (TiO_2_), silicon dioxide (SiO_2_), iron oxide (Fe_2_O_3_, Fe_3_O_4_), zinc oxide (ZnO), nickel oxide (NiO), gallium oxide (Ga_2_O_3_), and copper oxide (CuO), have been synthesized in different morphologies such as spherical, nanowires, triangular, nanotubes, and nanorods. With the reduced size and high-density corner or edge surface, oxide NPs display unique physical and chemical properties [87,88]. Their advantages include high stability; ease of preparation; customization to the desired size, shape, and porosity; swelling consistency; flexibility to incorporate into hydrophobic and hydrophilic systems; and ease of functionalization by various molecules due to the negatively or positively charged surface. Generally, metal oxide NPs are ionic particles, and the deposition of metal oxide NPs on the desired position with nanoscale resolution has a great potential for various applications in chemistry, electronics, optics, and biomedicine [89]. The main use of metal oxide NPs in electrochemical and biosensing applications is to strengthen the conductive sensing interface based on their catalytic property that allows electrical contact with the transducer surface. They can also work as electronic wires by facilitating fast electron transfer between the transducer and analyte molecule. Biocompatible metal oxide NPs have been used to fabricate immunosensors, enzyme sensors, and DNA sensors, while semiconducting NPs are primarily used as electrochemical markers and tracers. Nanoparticulated metal oxide can be coupled to the working electrode surface by several techniques such as physical adsorption, electrical deposition, chemical covalent bonding, and electro-polymerization [90]. Nanoparticulated metal oxide has been used in various applications, including small molecule electroanalysis, enzyme-based sensors, and immuno-and geno-sensors which are mainly used for cytokine detection. Genosensors utilize immobilized single-stranded DNA fragments on the electrode surface, while immunosensors are based on antigen–antibody interaction mostly in combination with electrochemical transduction. The performance of immunosensors depends on the density of immobilized antibodies/antigens, which is enabled by metal oxide nanostructures [89]. A few selected metal oxide NPs-based biosensors for cytokine detection are highlighted in this section. Key properties of these metal oxide NPs and their applications for cytokine sensing are summarized in Figure 4.

• Nanostructured zinc oxide (ZnO)

Recently, much attention has been given to the research on ZnO-based biosensors. Due to the distinctive electrical transport, high isoelectric point (IEP), low price, and high chemical stability, ZnO is a widely accepted candidate for biosensor applications [91]. Its high IEP allows better adsorption of enzymes, proteins, and DNA by electrostatic interactions. Its unique physicochemical properties have been exploited in a number of detection methods including electrochemical and optical methods. ZnO is known to be nontoxic and compatible with human skin, making it a promising material for use as a permanent biosensor in chronic diseases [91,92]. In the context of cytokine detection, some recent studies have shown good performance of biosensors based on ZnO nanorods. The ZnO nanorods allow enhanced fluorescence detection of DNA and protein samples and are therefore extensively investigated as an optical platform. The most recent development in this area is the multiplexed sensor for two biomarkers, TNF-α and IL-8, in urine to identify patients with the risk of having acute kidney injury. As compared to commercial ELISA-based sensors which have the detection limit of 5.5 pg/mL for TNF-α and 7.5 pg/mL for IL-8, the ZnO nanorod sensors have a lower detection limit of 4.2 fg/mL for TNF-α and 5.5 fg/mL for IL-8. In general, the ZnO nanorod sensors can detect TNF-α at the level well below the detection limit of ELISA. In addition to its superior sensitivity, the ZnO nanorod sensors offer the advantages of rapid analysis, minimal volume requirement, and reusability. The whole detection process can be completed within 90 min using only 60 µL of total sample volume. Furthermore, the biocompatible ZnO nanorod can survive at least 25 repeated assays with complex biological and chemical environments such as urine samples [17,93].

• Iron Oxide Nanoparticles

Iron oxide NPs including hematite (Fe_2_O_3_) and magnetite (Fe_3_O_4_) NPs have gained enormous attraction in biosensing due to their special magnetic, optical, and electrical properties. Their biocompatibility and durability make them highly desirable in many biomedical applications [94,95]. However, only a handful of iron oxide NPs-based biosensors have been reported, i.e., for the detection of IL-3 and IL-6. One of these studies utilized a nanocomposite of complexed longitudinal zeolite and iron oxide to improve surface characteristics of the biosensor designed for IL-3-mediated sepsis monitoring. The surface of zeolite-/iron oxide electrode were coupled to IL-3 antibody through an amine linker. The obtained sensor demonstrated the limit of IL-3 detection of 3 pg/mL due to the high degree of antibody immobilization on the sensing electrode [96]. Another study reported a new assay called “OnCELISA” for highly sensitive detection of trace cytokines secreted from individual live cells. This assay is the extension of ELISA by using cell surface to capture the secreted molecules that can be detected by fluorescent magnetic iron oxide NPs called Dragon Green superparamagnetic iron oxide (DG SPIO). This technique enhances the sensitivity of detection to 0.1 pg/mL, which is 10-fold more sensitive than standard fluorescence technique. This assay was implemented by surface biotinylation of cells followed by the attachment of IL-6 capture antibody. This secreted cytokine can then be captured on the cell surface immediately after their release instead of being diluted in the media. The captured cytokines are then labeled by DG SPIO-conjugated IL-6 detection antibodies and the fluorescence signals indicate the cytokine level. The labeling is stable after 12 h at 4 °C, and cell secretion and proliferation are not affected. However, the two antibodies required for OnCELISA should be produced against different epitopes of the same target cytokine [97]. 

• Titanium Dioxide (TiO_2_) Nanoparticles

TiO_2_ is a semiconductor material used in many applications such as photocatalysis, biosensors, and energy storage due to its properties such as high chemical stability, morphological versatility and biocompatibility. The wide applicability of recently developed TiO_2_ composites has accelerated the development and miniaturization of biosensing devices. The electrical characteristics of these nanostructures can be easily modified using structured shape and geometry to give higher surface reactivity and strong adsorption of biomolecules [98,99]. Sensitive nanostructured TiO_2_-based electrochemical biosensors for IL-6 detection have been developed based on surface-modified layers of TiO_2_ nanotubes. The nanotube layers are functionalized on the surface by direct immobilization of biological reagents. Strong antibody–antigen interaction has been used in two types of electrochemical biosensors: the impedimetric and amperometric. The impedimetric biosensor measures changes in capacitance caused by the formation of IL-6 antibody-antigen complex on the biosensor. For the amperometric biosensor, horseradish peroxidase (HRP) conjugated antibodies were used. The two electrochemical methods were reported to detect 5 pg/mL of IL-6, which is below the detection limit of commercially available ELISA kits [100]. The same technique was elaborated by the same research group to create an inexpensive, easy to prepare, and label-free cytokine array for the selective detection of IL-6, IL-8, and TNF-α. By immobilizing antibodies onto the TiO_2_ nanotube array via physical adsorption, IL-6, IL-8, and TNFα can be quantified simultaneously using electrochemical impedance spectroscopy (EIS). This impedimetric immunosensing technique shows good selectivity and high sensitivity against all analytes at the lower concentration than the normal concentration of these cytokines in the blood [101].

• Cerium Dioxide (CeO_2_) Nanoparticles

CeO_2_ is a nanostructured metal oxide commonly used in the biosensors field due to its easiness to immobilize enzymes or proteins on the electrode surface and its tremendous catalytic activity. In particular, immobilization of antibodies is crucial for the high sensitivity of the immunosensor [84]. CeO_2_ belongs to lanthanide metal oxide family and exhibits an excellent catalytic property due to the interconversion of oxidation states between Ce(III) and Ce(IV). In CeO_2_ nanostructures, Ce binds to oxygen to form fluorite crystalline structure. The transition of Ce(IV) to Ce(III) creates oxygen vacancies in the lattice. The presence of oxygen vacancies or defects in the lattice of each oxidation state improve electrocatalytic phenomena. Moreover, the high isoelectric point facilitates a strong binding interaction with biomolecules of low IEP via electrostatic interactions [102]. 

An ultrasensitive electrochemical immunosensor using Prussian Blue (PB) functionalized CeO_2_ NPs (PB-CeO_2_) has been developed for the detection of TNF-α. Preparation of PB-CeO_2_ started by coating CeO_2_ with positive charged chitosan (CS) layer. Negatively charged PB was then adsorbed on the CS layer. The synthesized NPs have a high sensitivity toward H_2_O_2_ detection. The electrochemical immunosensor was constructed on a conventional sandwich platform with immobilized primary anti-TNF-α antibody. Secondary anti-TNF-αantibody was crosslinked with PB-CeO_2_ through glutaraldehyde. The immunosensor was reported to have a low detection limit (2 pg/mL) for TNF-α over a wide linear range (0.005–5 ng/mL) [103]. CeO_2_ has also been used to improve the detection limit of ELISA-based cytokine assay kits together with magnetic Fe_3_O_4_ particles. Based on this principle, a magnetic colorimetric assay for IL-6 detection was developed by the immobilization of the capture antibody on magnetic Fe_3_O_4_ particles. The secondary antibody was coupled on CeO_2_ spheres. The labelled CeO_2_ catalyzed the oxidation of o-phenylenediamine to 2,3-diaminophenazine, a yellow product with absorption wavelength at 448 nm. The assay demonstrated the detection limit of IL-6 at 0.04 pg/mL with the linearity range of 0.0001–10 ng/mL [104].

• Silicon Dioxide (SiO_2_) Nanoparticles

SiO_2_, or silica, is a covalently bonded material consisting of silicon and oxygen. It is a component of sand that can be found naturally as quartz. SiO_2_ is usually white or colorless and not water-soluble. SiO_2_ NPs are important material for the optical fibers production and have been used for the construction of nanobiosensors due to their biocompatibility and good electron transfer properties. Their inherent resonance light scattering (RLS) characteristic can be used to develop a simple sensing method. SiO_2_ NPs possess a porous structure, high thermal resistance, and very high surface activity and adsorption properties. The structural features (e.g., size, shape, and porosity) of SiO_2_ NPs can be facilely modified to improve their performance. The silane chemistry enables surface engineering of SiO_2_ NPs with diverse molecules and polymers. Such surface functionalization was shown to influence the biological activities of SiO_2_ NPs in vivo, including blood circulation, biodistribution, and cellular internalization [105]. The refractive index of SiO_2_ NPs is similar to that of polymeric coatings, so it finds several applications in nanocomposites that improve the sensitivity and specificity of detection, and reduce the toxicity of biosensors, i.e., by encapsulation or sandwich formation [106,107,108,109,110]. To date, SiO_2_ NPs have been used in cytokine biosensors as an immobilization substrate or signal amplifier. Lee et al. developed a novel substrate based on hafnium oxide (HfO_2_) grown on SiO_2_ for IL-10 detection. The sensor measured IL-10 within the linear range of 0.1–20 pg/mL and sensitivity of 0.49 ng/mL [111]. As a signal amplifier for cytokines detection, QD-polymer-functionalized SiO_2_ nanospheres have been used for TNF-α measurement. Anti-TNF-α antibody was bonded to the functionalized nanospheres to obtain QD-polymer-functionalized SiO_2_ probes which were attached to the electrode surface. Enhanced sensitivity of detection can be accomplished by increasing the QD loading per immunoassay. With this approach, 10-and 5.5-fold increase in detection sensitivity from electrochemiluminescence (ECL) and square-wave voltammetry (SWV) measurements compared to unamplified method were observed, respectively. The detection limits of 7 and 3 pg/mL were achieved for the ECL and SWV detection of TNF-α, respectively [112]. 

### 5.4. Carbon-Based Nanomaterials

Due to their exceptional optical, electrical, mechanical, chemical, and thermal properties, extensive research has been carried out on carbon-based nanomaterials. Carbon is commonly used for the construction of electrodes based on its unique electrochemical properties, e.g., small background current and large potential window, as well as its low cost. Moreover, carbon nanomaterials are biocompatible. To date, different carbon allotropes such as graphene, carbon nanotubes (CNTs), fullerenes, carbon dots, carbon black, and nano-diamonds have been exploited to achieve specific properties or desirable features of biosensors [113,114,115]. Developed cytokine biosensors based on carbon nanomaterial are summarized in Table 3.

• Graphene

Graphene is the most prominent member of carbon nanomaterials for biosensor applications. The physical, optical, and electrochemical advantages of graphene have led to improved sensitivity and selectivity of biosensors [116]. However, there are some problems associated with graphene-based biosensors. For instance, graphene exhibits a high agglomeration tendency, making it difficult to prepare, and there are no standardized procedures for its biosensors. Moreover, the physical stability of graphene in biological media, as well as its potential toxicity, remain a concern [117,118]. Graphene-based multi-analyte electrochemical immunoassays have been developed for rapid and ultrasensitive detection of IL-6 and matrix metallopeptidase-9 (MMP-9) simultaneously. The sensor utilized a graphene nanoribbon (GNR)-modified electrode to immobilize the antibody and amplify the electrochemical signal. With sandwich-type immunoassay strategy, simultaneous detection without crosstalk between the sensing electrodes was achieved. The detection range of 10^−5^ to 10^3^ ng/Ml with the detection limits of 5fg/mL for MMP-9 and 0.1pg/mL for IL-6 reported [119]. Several other studies investigated the benefit of graphene as a label-free sensor for IFN-γ. Ruecha et al. [120] reported a novel paper-based microfluidic device as a label-free electrochemical impedance immunosensor. The electrode was fabricated by depositing polyaniline (PANI) on a graphene screen-printed paper electrode. This modified electrode exhibited high electrochemical conductivity and surface area resulted in significant improved in sensor sensitivity due to better antibody immobilization. The developed sensor can detect IFN-γ at the lowest concentration of 3.4 pg/mL with linear relationship between the impedance and IFN-γ concentrations in the range of 5–1000 pg/mL [120]. In another study, IFN-γ aptamer was immobilized on a graphene surface to create a field-effect transistor (FET) sensor that measures the change in conductance due to binding of the molecules of interest. The graphene structure was incorporated onto a PDMS substrate. In the presence of analytes, a change in charge distribution occurs, which results in an increase in electron transfer that can be monitored via current measurements. Such current changes were found to correlate with IFN-γ concentrations in the nano to micromolar range with the detection limit of 83 pM [121].

• Nanodiamonds (NDs)

NDs are nanomaterials with the crystal structure and specific properties of diamond. They are small, but have large surface area. Their high adsorption capacity allows attachment of various chemical and biological molecules to their surface. NDs have low toxicity, with exceptional hardness, refractive index, thermal conductivity, chemical stability, and coefficient of friction. Due to the presence of a complex defect N-V, containing nitrogen (N) and a vacancy (V), they can also emit fluorescence signal which can be used for several applications [114,122]. Stochastic ND-based sensors for the detection of IL-1β, IL-6 and IL-12 have been reported. By creating nano-sized pores in a membrane, ionic transport through the pore in the presence of analytes was measured by the sensor. The magnitude, duration, and rates of occurrence of the resulting current blockade allow rapid analysis of analyte concentrations, as well as discrimination of the similar molecular species [123]. Reliable determination of the cytokine concentrations was demonstrated in whole blood, urine, and brain tumor samples [124]. NDs have also been explored for the application as insertable optical fibers for in vivo biosensing of cytokines. Captured antibodies were immobilized on the optic fiber surface. The functionalized fiber was placed in vivo for cytokine capturing, and it has successfully been used for the detection of IL-6. The sensor demonstrated the sensitivity of 0.4 pg/mL with spatial resolution of 200 µm. It has also been used to measure IL-1β release in discrete brain regions in animals via inserted fiber optic device [125,126].

• Fullerenes

Fullerenes are carbon-based hollow nanomaterials. The structure characteristic of fullerenes is the establishment of atomic C_n_ clusters (n > 20) on a spherical surface. C_60_ is the most extensively studied fullerene. It is a symmetric spherical molecule consist of 60 carbon atoms with the diameter of 0.7 nm. C_60_ offers several advantages, such as high symmetry, low cost, ease of production, inertness (under mild conditions), and non-toxicity. Fullerenes have unusual properties, such as multiple redox states, stability in different redox forms, and light source switching. They are also easy to be functionalized. Therefore, they have been used in many different applications, including superconductors, biosensors, catalysts, and optical and electronic devices [127,128]. Fullerenes have been used to increase the sensitivity of commercially available carbon electrodes for the detection of IL-8 via silver stripping voltammetry technique. The analyte concentration can be measured through current measurements over the scanned potential. The analyte is pre-concentrated at the electrode and then stripped by applying a potential. The nanostructured electrodes showed higher sensitivity than bare carbon electrodes with the detection limit of 0.61 ng/mL [129]. A nanocomposite containing fullerene and multi-walled carbon nanotubes (MWCNTs) was used to create a sensitive label-free electrochemical immunosensor for TNF-α detection. TNF-α antibodies were entrapped onto the fullerene-MWCNT nanocomposite and used as a sensing platform and electrocatalyst for oxidation. TNF-α was determined based on its obstruction of the electrocatalytic oxidation upon the binding to the electrode surface via an interaction with TNF-α antibody. Under optimal conditions, the electrochemical immunosensor can detect TNF-α with the dynamic range of 5–75 pg/mL and detection limit of 2.0 pg/mL [130].

• Carbon Nanotubes (CNTs)

CNTs are basically one layer of graphene rolled into a hollow tube with a diameter in nanoscale. Based on their structures, they can be simply classified into two groups: single-walled (SW) and multi-walled (MW) CNTs. Compared to other fibrous materials, CNTs possess excellent physical properties such as rigidity, tensile strength, and elasticity. They exhibit high aspect ratio and high thermal and electrical conductivities. CNTs have spurred much interdisciplinary research due to their unique structural, electronic, optoelectronic, mechanical, and chemical properties. CNT-based materials have increasingly been recognized as the next generation building block for ultra-sensitive and ultra-fast biosensors. Their strong asset is their constitute scaffolds, which accommodate functionalization by several recognition and signal transduction entities, thus allowing multiplexed testing of analytes. With greater ability to conduct electricity than copper wires, CNTs can potentially be a good material for the transduction of electric signals generated upon the biorecognition event of a target [131,132]. 

CNTs have been applied for the detection of several cytokines including IL-6, TGF-β, and IL-1β. Enzyme-labeled immunosensors for sensitive IL-6 detection have been developed based on dual amplification of AuNPs and CNTs. In this platform, the capture antibody was immobilized on the AuNP-modified electrode, while the detection antibody, which is conjugated with HRP, was coupled to CNTs for signal detection in the presence of H_2_O_2_. The system showed a linear response range of 4–800 pg/mL with the detection limit of 1 pg/mL for IL-6, consistent with standard ELISA results, demonstrating its reproducibility, selectivity, and reliability [133]. SWCNTs have been used as a nanotag in an amperometric biosensor for TGF-β. In this approach, the nanotubes were hybridized with viologen, an organic compound that can reversibly change its color through reduction and oxidation. The TGF-β antibody with HRP tag was then covalently linked to the viologen–SWCNT hybrid. Capture antibody was immobilized on the carbon electrodes and sandwich-type immunoassay was employed with signal amplification from the viologen–SWCNT-antibody-HRP as a carrier tag. The immunosensor yielded a linear detection range of 2.5–1000 pg/mL and detection limit of 0.95 pg/mL, which is better than other immunosensors or ELISA kits for TGF-β detection. Such high sensitivity was also observed with other proteins. Validation of the sensor was performed in saliva samples. Minimal sample preparation was required, and the results obtained were in good agreement with standard ELISA results [134]. 

In another study, an ultrasensitive electrochemical immunosensor based on CNTs forest electrodes was reported. SWCNT forests with captured IL-6 antibodies were used in an electrochemical sandwich immunoassay. The detection antibody was attached to HRP-labeled MWCNTs. This setup provided the most sensitive detection of IL-6 with the detection sensitivity of 0.5 pg/mL (25 fM) [135]. MWCNTs have also been used to fabricate electrochemical sensors for IL-1β detection using electro-click chemistry to immobilize ethynylated antibody onto azide-functionalized MWCNTs. The sensor was then used to perform sandwich-type immunoassay with biotinylated anti-IL-1β as a detection antibody. The electrochemical immunosensor exhibited two linear ranges from 10–200 pg/mL and 200–1200 pg/mL with the detection limit of 5.2 pg/mL, a significant improvement over standard ELISA assay. Outstanding performance of the sensor was observed when the sensor was used to determine the amount of IL-1β in saliva without the need of sample pre-treatment [136]. There has been an attempt to coat CNTs with cadmium telluride (CdTe) QDs and a single molecule of captured antibody to improve IL-6 detection. Sample IL-6 competed with surface-bound IL-6 for the binding sites on the captured antibody, which led to the depletion of CNT-QD-antibody on the sensor surface with increasing sample IL-6 concentrations [137]. 

CNTs have also been used for label-free impedimetric immunosensor development. Impedimetric immunosensors consist of a transducing electrode coated with a biorecognition element (antibody or aptamer) where binding of the target protein results in a change in the electrical impedance. There was a study to develop an impedimetric immunosensor for IL-6 detection by integrating SWCNT and AuNPs. The sensing platform was constructed by depositing AuNPs on SWCNT array on SiO_2_/Si substrate. The developed sensor can detect IL-6 in serum at a very low limit of 0.01 fg/mL, which is much lower than that the results from the sensor using CNTs forest or AuNPs alone. The combination sensor demonstrated wide linear response from 0.01–100 fg/mL with a low response to several interferences found in serum. The sensor also possesses a high stability with no change in sensitivity after being stored for a month at 4 °C [138]. Another example of label-free cytokines sensors was reported by Khosravi et al. [139] who employed aptamer-functionalized CNTs with aptamer as the recognition element. In this platform, SWCNTs were functionalized with 1-pyrenebutanoic acid succinimidyl ester (PASE) conjugated IL-6 aptamers. In the presence of IL-6, the conductivity of functionalized CNTs was decreased in the correlation to the concentrations of IL-6 [139]. 

### 5.5. Polymer Nanomaterials

Polymeric nanomaterial is a common term used for all polymer-based nanomaterials. They are obtained from synthetic or natural sources and can be classified based on their in vivo behavior as biodegradable or non-biodegradable. Polymer nanocomposites have a large surface area, high electrical conductivity, and fast electron transfer rate which make them a good transducer in electrochemical sensors. They can be synthesized with several nanoscale variations of fillers leading to various specific sensing applications. The fillers enable ion diffusion by intercalating into the polymer matrices resulting in improved stability of active electron transfer and detection limit of the sensor [140]. To date, several polymeric nanomaterials, including conducting polymers (CPs), dendrimers, and molecular-imprinted polymers (MIPs), have been developed for the detection of various analytes [84]. 

CPs are polyconjugated polymers consist of alternating single and double bonds. With this molecular structure, CPs exhibit electrical and optical properties that are suitable to be used as sensing transducer. Inclusion of CP nanostructures in the sensor transducer has been shown to improve sensing performance significantly [141]. Several CPs have been integrated into electrochemical immunosensors, including polypyrrole (PPy), PANI, poly-2,5-di-(2-thienyl(-1-pyrrole-1-(p-benzoic acid) (DPB), poly-(3,4-ethylene dioxythiophene) (PEDOT), and poly-pyrrolepropionic acid (PPA). PPy, a five-membered heterocyclic ring, is prominent for its tunable conductivity, ease of synthesis, reversible redox activity, and environmental stability. As PPy nanostructures can be prepared in both aqueous and non-aqueous solutions through electrochemical or chemical interaction, they are particularly useful for various applications, including biosensors [141,142]. Aptamer probes doped in PPy have been employed for the detection of platelet-derived growth factor (PDGF), an important inflammatory cytokine, by EIS with the reported sensitivity of 10 ng/mL [143]. PPy-modified silicon nitride (Si_3_N_4_) substrate has similarly been employed to develop a label-free capacitance impedimetric immunosensor for IL-10 detection. The immunosensor was created by chemically depositing the PPy conducting layer on Si/SiO_2_/Si_3_N_4_ substrate, followed by IL-10 antibody immobilization, and detection by EIS. The developed sensor showed a linear detection response in the range of 1–50 pg/mL with a high sensitivity of 0.1128 pg/mL and very low detection limit of 0.347 pg/mL [144].

PANI is a semiconducting polymer that contains repeating units of benzenoid diamine and quinoid diamine. PANI has high surface to volume ratio, as well as distinct redox couples based on its repeating units, resulting in effective electron transfer. To date, 2D PANI is one of the promising nanostructures in flexible biosensors development. However, diminished conductivity of PANI under physiological conditions tends to be an obstacle for its application. PANI has been employed in several studies to develop biosensors for IFN-γ. One study used magnetic NPs coated with PANI and captured antibody to magnetically capture IFN-γ from the sample. AuNPs conjugated with detection antibody and cadmium sulfide (CdS) NPs were then added to form a multifunctional complex and added on a screen-printed carbon electrode (SPCE) chip. The electrochemical signal from CdS-NPs was then measured by stripping voltammetry. The whole detection process with the developed immunosensor can be completed in about 1 h [145]. In another study, a screen-printed paper electrode (SPPE) was used to detect IFN-γ by microfluidic immunosensors. The electrode consists of graphene coated with PANI, which enables covalent immobilization of IFN-γ antibody. One unique advantage of this paper-based sensor is its ability to be mass produced at a low cost [120]. Another nanocomposite of graphene oxide, PANI and QDs has been used for electrochemiluminescence detection of IL-6 with the sensitivity range of 0.0005–10 ng/mL [146].

PEDOT is one of the most explored CPs. It is polythiophene derivative synthesized from 3,4-ethylenedioxythiophene (EDOT) monomer. It has exceptional conductivity, good optical properties, and electronic stability in physiological settings. Currently, the most studied PEDOT derivative is PEDOT:polystyrene sulfonate (PSS). It is the co-polymer mixture of PEDOT and sodium polystyrene sulfonate. PEDOT:PSS exhibits high conductivity and stable in oxidized state. It is also visible light transmissive and water dispersible. To date, PEDOT:PSS is commonly used as a flexible polymeric electrode based on its flexibility, tunable conductivity, biocompatibility, and low cost [141,147]. 

MIP-based biosensors have been investigated for the detection of various molecules. Although imprinting large or complex structures such as proteins is still a challenge, imprinting techniques is well demonstrated to reveal selective organization of functional groups in the imprinting biomolecules. Nanostructured MIPs can be produced by a variety of methods including dispersion, suspension, and emulsion-seeded polymerization [84]. A combination of MIP and PEDOT has been used for the detection of IL-1β via an electrochemical MIP sensor. To enhance the detection sensitivity and signal stability, the carbon backbone was functionalized with PEDOT/4-aminothiophenol and MIP film. Using EIS detection, a linear response range of 60 pM to 600 nM with a detection limit of 1.5 pM was obtained. Selectivity tests confirmed good selectivity of the developed biosensor [148].

### 5.6. Bionanomaterials

Bionanomaterials are biomaterials with a nanoscale dimension that comprise partially or totally of biological molecules such as antibodies, proteins, DNA, RNA, polysaccharides, and lipids. Numerous efforts have been made on the development of hybrid structures containing bionanomaterials and other functional NPs (e.g., CNTs, graphene, nanometal, and QDs) for applications in biosensors, bio-imaging, biomineralization, biocatalysts, and drug delivery [84]. The most widely used bionanomaterial for cytokines biosensors is aptamer. Aptamers are single-stranded oligonucleotides (DNA or RNA) that can form high-affinity complexes with their targets via specific conformational changes and can be used as an alternative small biomolecule recognition element [149]. Several aptamers have been incorporated in cytokine biosensors, e.g., fluorescence DNA/RNA probes. In these aptamer-based sensors, the release of quencher-labeled complementary strand or fluorophore-labeled aptamer upon the binding of analytes is detected [17]. Few studies reported IFN-γ detection using Förster resonance energy transfer (FRET)-based aptasensors. An aptamer variant was constructed by the hybridization of fluorophore-labeled aptamer with quencher-labeled complementary oligonucleotide strand. IFN-γ displaced the quenching strands, disrupting FRET and resulting in fluorescence signal. The signal can be observed directly upon binding of the IFN-γ. This aptasensor was reported to be able to detect IFN-γ in a linear range from 5 to 100 nM with the detection limit of 5 nM [150]. The sensor was further developed for real-time monitoring of cytokine secretion. Micro-patterned aptamer beacons were fabricated by micro-patterning glass substrate with polyethylene glycol (PEG) hydrogel microwells to define sites for attachment of aptamer molecules. In the presence of IFN-γ, the quencher displaced from fluorophore-labeled aptamer and fluorescence signal in the wells increased. The real-time changes in fluorescence signal was confined to microwells and depends on IFN-γ concentration [151].

In this section, some interesting applications of nanomaterial for the development of cytokine biosensors are presented. In general, several parameters should be considered before deciding which nanomaterials to use, including dynamic range, sensitivity, cost, ease of miniaturization, and biocompatibility. An increasing number of studies regarding the combination of these nanomaterials to produce new nanocomposites with enhanced performances have been reported in the literature. Table 4 compares the strengths and weaknesses of different nanomaterials for cytokine biosensor application. To date, there is no commercially available cytokine biosensors for POC application. Most of the discussed studies are still in the developmental stage at academic research laboratories. However, integration into real application has been accelerated, i.e., a wearable SWEATSENSER device for the detection of IL-6, IL-8, IL-10, and TNF-α fabricated with ZnO nanofilm. The device demonstrated analytical range of 0.2–200 pg/mL and had been shown in a clinical study to be able to distinguish healthy subjects from subjects with infection [152].

## 6. Conclusions and Future Perspectives

Advanced cytokine biosensors are being developed at a rapid pace due to the increasing realization of the crucial role of cytokines in disease pathogenesis and the need for early diagnosis, disease monitoring, and therapeutic response assessment. This article reviews the evolving nanomaterials-based biosensors for cytokine detection, as well as the strategies and limitations associated with these biosensors. Major signal transduction methods employed in nanomaterials-based biosensors including optical, electrochemical, and magnetic transduction methods were mentioned. Nanomaterials discussed are QDs, noble metal nanomaterials, metal oxide nanomaterials, carbon-based nanomaterials, and bionanomaterials. Several of these biosensors utilize hybrid nanomaterials to achieve the specific application needs. Examples of these biosensors and their integrated nanomaterials are described, along with their applications in cytokine measurements.

Most cytokines in the body function in concert with other cytokines to elicit their biological or pathological effects. Consequently, there is a need for multiplexed detection systems that can detect multiple cytokines simultaneously. Several multiplexed cytokine biosensors have been developed in recent years, largely through the incorporation of new nanomaterials and surface coating or functionalization of the nanomaterials with multiple biosensing moieties. Due to their increasing complexity, the issues of portability and ease of use should be carefully addressed to promote their applications. Hardware and software packages will also need to be developed to handle the complexities of future devices and to provide user-friendly readouts. Advances in nanomaterials science and bioengineering have paved the way for ultrasensitive detection and real-time monitoring of multiple cytokines that could not be achieved by conventional detection methods. Further development of these technologies will enable better and more sensitive cytokine biosensors. However, the current implementation of these technologies has been limited to some specialized research fields and analytical laboratories. Translation of these biosensor technologies into clinical and hospital settings remains a challenge. For POC applications, portability, sensitivity, specificity, stability, and real-time monitoring are critical. Wearable devices can be an exciting opportunity in healthcare due to their potential to change the status quo in patient care, with tools that the patients can use to track their own conditions and take responsibility for their own health. The unique properties of nanomaterials, such as high conductivity, large aspect ratio, high tensile strength, and fast electron transfer rate, make these materials well suited for wearable biosensor applications. 

Currently, the information on large-scale fabrication of nanomaterial-based devices is still lacking, so their practical applications have not yet been fully realized. Power supply is another important issue in wearable sensors. With the trend toward miniaturization and flexibility to function at biological interfaces which may be soft, folded, or curvy, rigid batteries would be ineffective. Battery-free and wireless sensing systems, such as near-field communication (NFC), could be a good choice for future research in this field. NFC modules allow real time and wireless data transmission without the need for battery. One limitation of this technology is its short transmission range, which results in the need for a close distance between the sensor and data receiving unit. 

Despite their many advantages, nanomaterials do pose human health risks and potential threat to the environment. Nanoscale materials, due to their exceptionally small size, can enter the systemic circulation and penetrate deeper tissues to cause unintended health effects. To date, several nanomaterials have been investigated for their biocompatibility and biodegradability to avoid such consequences. These studies should be conducted in parallel with their performance assessment to ensure safe and effective utilization of the technologies.

## Figures and Tables

**Figure 1 biosensors-11-00364-f001:**
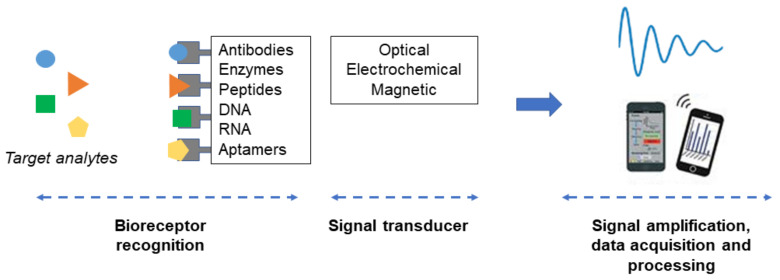
Schematic diagram showing biosensor components: bioreceptor recognition, transducer, and data acquisition.

**Figure 2 biosensors-11-00364-f002:**
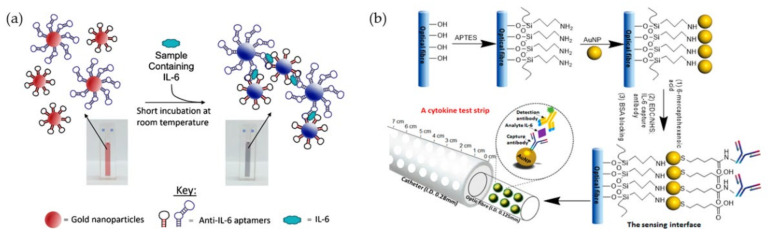
Illustration for some examples of the developed AuNPs-based cytokine biosensors. (**a**) A schematic of the aptamer-AuNPs-based aggregation assay for the IL-6 detection. Reproduced with permission from Ref [70]. Copyright 2019, Giorgi-Coll et al. (**b**) Scheme of the cytokine test strip preparation based on the optical fiber for IL-6 detection. Reproduced with permission from Ref [73]. Copyright 2016, American Chemical Society.

**Figure 3 biosensors-11-00364-f003:**
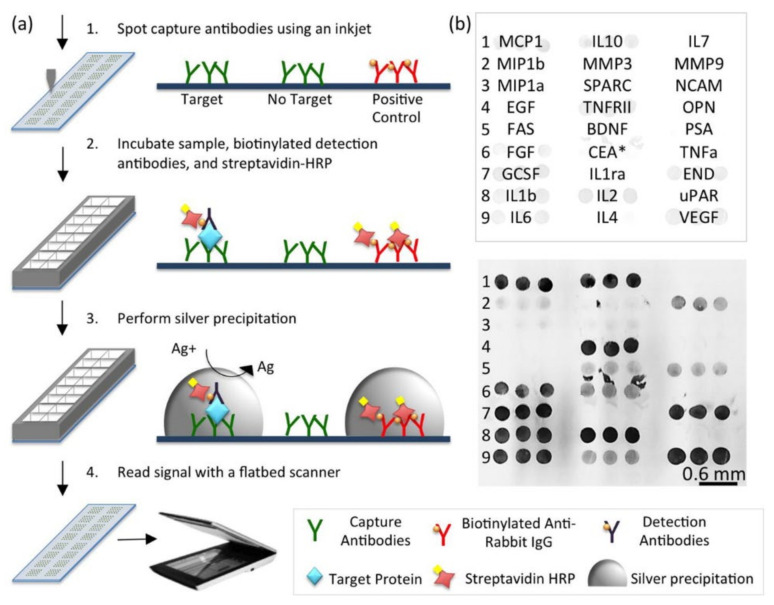
Schematic diagram of multiplex SENSIA. (**a**) Typical assay protocol. (**b**) Spotting pattern of capture antibodies against 27 proteins and an example assay result. Reproduced with permission from Ref [78]. Copyright 2015, American Chemical Society.

**Figure 4 biosensors-11-00364-f004:**
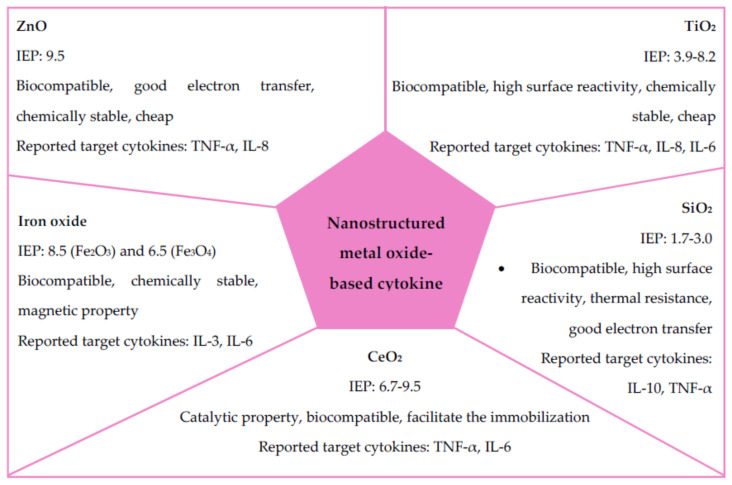
Representative metal oxide nanomaterials that have been used for the development of cytokines biosensors (IEP, isoelectric point).

**Table 1 biosensors-11-00364-t001:** Classification of cytokines based on structure and function subgroups (modified from [5]).

Cytokine Family	Functions	Example
*Interleukins*	-stimulate hematopoiesis	IL-3, IL-7
	-regulate pluripotency and inflammation	IL-1, IL-6
	-regulate T cells and B cells	IL-2, IL-4, IL-5, IL-12, and IL-13
*Interferons*	-exert antiviral, anti-proliferative effects	INF-α, INF-β
	-exert antitumor effect	INF-γ
*Tumor necrosis factors*	-stimulate inflammation, apoptosis, non-specific immune response	TNF-α, TNF-β
*Chemokines*	-regulate migration of granulocytes and lymphocytes, promote angiogenesis and inflammation	CXCL-1, CXCL-8
	-regulate migration of monocytes	CCL-3, CCL-5, CCL-7, and CCL-8
*Colony stimulating factors*	-stimulate proliferation and maturation of myeloid precursors	G-CSF, GM-CSF
*Transforming growth factor*	-stimulate fibroblast proliferation and extracellular matrix production	TGF-β

**Table 2 biosensors-11-00364-t002:** Advantages and disadvantages of the discussed conventional techniques for cytokine detection.

Method	Advantages	Disadvantages
*ELISA*	-well-accepted and standardized protocol -commercial kits available for a wide range of cytokines -high sensitivity and specificity	-laborious assay procedure -high cost
*RIA*	-high sensitivity	-exposure to radiations -time consuming procedure -costly equipment
*ELISPOT*	-detection of cytokines from single cells	-difficult to interpret the result
*qRT-PCR*	-sensitive and well-developed	-require proper handling of samples to avoid RNA degradation -amount of detected RNA may not correlate with protein level
*Immunostaining*	-provide physiologic and pathologic information	-can be invasive to obtain tissue of interest -time consuming procedure -fixation step can denature cytokines -need optimization for each antibody
*ICC*	-multiple cytokines can be detected -fast process compared to other methods	-procedure contains fixation step -costly equipment
*Microarray*	-multiplex	-require extensive validation

**Table 3 biosensors-11-00364-t003:** Recent studies on carbon nanomaterials-based cytokine biosensors mentioned in this review.

Allotrope	Sensor Platform/Label	Analyte	Detection Method	Linearity Range	LOD	Ref
*Graphene*	GNR/HSPCE/PS@PDOP	IL-6	Electrochemical	10^−5^–10^3^ mg/mL	0.1 pg/mL	[119]
	PANI/Graphene	IFN-γ	CV	5–1000 pg/mL	3.4 pg/mL	[120]
	Graphene/PDMS/aptamer	IFN-γ	Electrochemical	NA	83 pM	[121]
*NDs*	NDs/PIX	IL-1β, IL-6, IL-12	Stochastic electrochemical	IL-1β: 4 × 10^−9^–6.4 × 10^−5^ µg/mL, IL-6: 4 × 10^−9^–1 µg/mL, IL-12: 5.1 × 10^−7^–8 × 10^−3^ µg/mL	IL-1β: 4 × 10^−9^ µg/mL IL-6: 4 × 10^−9^ µg/mL IL-12: 5.1 × 10^−7^ µg/mL	[124]
	NDs optic fiber/magnetic NPs	IL-6	Fluorescence	0.4–400 pg/mL	0.1 pg/mL	[126]
*Fullerenes*	SPES/Fullerenes	IL-8	ASV	NA	0.61 g/mL	[129]
	Fullerenes/CNTs/Ionic liquid	TNF-α	Electrochemical	5–75 pg/mL	2 pg/mL	[130]
*CNTs*	AuNPs/PDOP/CNTs	IL-6	Amperometry	4–800 pg/mL	1 pg/mL	[133]
	HOOC-Phe-SWCNTs/Viologen/HRP	TGF-β	Electrochemical	2.5–1000 pg/mL	0.95 pg/mL	[134]
	CNTs/Iron oxide/HRP	IL-6	Amperometry	NA	0.5 pg/mL	[135]
	Cu catalyzed IgG/azide-MWCNTs	IL-1β	DPV	10–1200 pg/mL	5.2 pg/mL	[136]
	AuNPs/SWCNTs/SiO_2_	IL-6	EIS	0.01–100 fg/mL	0.01 fg/mL	[138]
	SWCNTs/aptamer-PASE complex	IL-6	Conductometry	NA	10 pg/mL	[139]

HSPCE = heated screen-printed carbon electrode, PS@PDOP = polydopamine-coated polystyrene, CV = cyclic voltammetry, PDMS = polymethylsiloxane, PIX = protoporphyrin IX, ASV = anodic stripping voltammetry, SPES = screen printed electrochemical sensors, DPV = differential pulse voltammetry, EIS = electrochemical impedance sensor, and NA = not available.

**Table 4 biosensors-11-00364-t004:** Comparison of important features of different groups of nanomaterials discussed in preceding paragraph for cytokine biosensors application.

Nanomaterials	Advantages	Disadvantages
QDs	High photostability Long lifetime	Toxicity in in vivo system Interact with protein in biological fluid
NMNs	Can be used in both biorecognition and signal amplification purposes Several transduction methods can be applied	Difficult to control morphology during synthesis Cost-ineffective for large scale production
Metal oxide	Charge and size controllable Fast response and recovery time Several transduction methods can be applied	Questionable toxicity for some of them Often need to be used with other nanomaterials as a hybrid nanocomposite for better sensitivity
Carbon-based	High degree of selectivity when functionalized Several transduction methods can be applied	Toxicity in in vivo system
Nanopolymer	Flexibility Wide range of polymers	Difficult to functionalize due to complex structure
Bionanomaterials	Superior specificity	Need other materials as transducer

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
