# Peer review of "Advances in Nanotechnology-Based Biosensing of Immunoregulatory Cytokines"

_biosensors, 2021, doi:10.3390/bios11100364_

Round 1

Reviewer 1 Report

In this review, the authors describe various biosensors/immunosensors for single or multiplexed ultrasensitive cytokine detection. High emphasis is made on a large variety of nanomaterials, their synthesis and biocompatibility are addressed, as well as progress in wearable or POC systems. Both biosensor and nanomaterial advantages and limitations are addressed. The authors made a great effort documenting and analyzing so many biosensor architectures and managed to write a well-organized, comprehensive review.

Overall, the manuscript is well written and very interesting; however, the reviewer has a few suggestions for improvement and there are a few specific requirements:

  • The introduction is completely dedicated to cytokines (description and function), where no reference to lab-on-a-chip or POC systems is made, although they are well discussed throughout the manuscript. Also, more emphasis on the various nanomaterials should be made since they are the essence of this review.
  • Chapter 2: It is nice to have such a comprehensive description of the classical techniques for cytokine detection, however, the reviewer feels that too much attention was given to this section. Also, the data in Table 2 overlaps with the description for each technique. The authors are asked to make sure that the same information is not duplicated in both text and table.
  • Line 224: correct word “detectiopn”
  • Line 308-309: revise sentence “Smaller analytes generally more diffusive and have lower steric hindrance thereby better detection efficiency can be achieved [42].”
  • Line 352: correct word “fluoresce”
  • Line 409: “There were micrometer-sized……”
  • Line 666: replace “milieu” with “media”, milieu refers to a person's social -environment
  • Line 785-786: revise sentence “A simple impedimetric immunosensor was developed by integrating two nanomaterials that previously been used for IL-6 detection”
  • Line 811 “that are suitable ….”
  • The authors should add a table for chapter 5: there is so much information, that it would be a great addition to the manuscript (table summarizing the biosensors with the various nanomaterials, the detected cytokine – single or multiple and the performances - limits of detection).

Author Response

Reviewer 1

In this review, the authors describe various biosensors/immunosensors for single or multiplexed ultrasensitive cytokine detection. High emphasis is made on a large variety of nanomaterials, their synthesis and biocompatibility are addressed, as well as progress in wearable or POC systems. Both biosensor and nanomaterial advantages and limitations are addressed. The authors made a great effort documenting and analyzing so many biosensor architectures and managed to write a well-organized, comprehensive review.

Overall, the manuscript is well written and very interesting; however, the reviewer has a few suggestions for improvement and there are a few specific requirements:

  • The introduction is completely dedicated to cytokines (description and function), where no reference to lab-on-a-chip or POC systems is made, although they are well discussed throughout the manuscript. Also, more emphasis on the various nanomaterials should be made since they are the essence of this review.

Thanks for the suggestion. We have now mentioned POC in the introduction. More introduction on nanomaterials has been added as well. Please see the introduction section: Page 2, line 62-70; Page 3, line 71-73, 78-82 and 84-86.

  • Chapter 2: It is nice to have such a comprehensive description of the classical techniques for cytokine detection, however, the reviewer feels that too much attention was given to this section. Also, the data in Table 2 overlaps with the description for each technique. The authors are asked to make sure that the same information is not duplicated in both text and table.

Chapter 2 has been shortened and the overlapped text have been removed according to the reviewer’s comment.

  • Line 224: correct word “detectiopn”

It has been changed to “detection”. It is now on Page 7, line 227.

  • Line 308-309: revise sentence “Smaller analytes generally more diffusive and have lower steric hindrance thereby better detection efficiency can be achieved [42].”

The sentence has been revised. Please see Page 9, line 311-312.

  • Line 352: correct word “fluoresce”

We checked the spelling carefully and the word is correct as verb.

  • Line 409: “There were micrometer-sized……”

It has been changed according to the suggestion. Please see Page 11, line 412.

  • Line 666: replace “milieu” with “media”, milieu refers to a person's social environment

The word has been changed according to the suggestion. Please see Page 18, line 713-714.

  • Line 785-786: revise sentence “A simple impedimetric immunosensor was developed by integrating two nanomaterials that previously been used for IL-6 detection”

The sentence has been revised. Please see Page 21, line 832-833.

  • Line 811 “that are suitable ….”

It has been changed according to the suggestion. Please see Page 22, line 863.

  • The authors should add a table for chapter 5: there is so much information, that it would be a great addition to the manuscript (table summarizing the biosensors with the various nanomaterials, the detected cytokine – single or multiple and the performances - limits of detection).

We agree that there is quite a lot of text in this section. So, we added 2 figures and 2 tables to this section. Figure 2 (Page 12) and Figure 3 (Page 13) are illustration diagrams of some biosensors from AuNPs and AgNPs described in the article. Table 3 (Page 21-22) is the summary of carbon-based cytokine biosensor discussed in the review. Table 4 contains information on the important features that are considered to be the strength and weakness of each nanomaterials in biosensor applications (Page 24-25).

Reviewer 2 Report

Lohcharoenkal and her co-workers presented in this review the "Advances in nanotechnology-based biosensing of immunoregulatory cytokines". The manuscript, in general, is well written, The topic is of great interest.    Further comments:   --Sensor performances, such as linearity range, limit of detection (LOD), limit of quantification (LOQ), repeatability, intermediate precision, selectivity, and lifetime, should be discussed in the review. Authors should compare the performance of different sensors .    --Authors described graphene, fullerenes, CNTs etc. as good candidates for biosensor applications. Authors should cite recent relevant reviews there.  Recent Advances in Sensing Applications of Graphene Assemblies and Their Composites. Adv. Funct. Mater. 2017, 27, 1702891. Doi:10.1002/adfm.201702891; and Carbon Nanomaterial Based Biosensors for Non-Invasive Detection of Cancer and Disease Biomarkers for Clinical Diagnosis. Sensors 2017, 17, 1919; DOI:10.3390/s17081919.

Author Response

Lohcharoenkal and her co-workers presented in this review the "Advances in nanotechnology-based biosensing of immunoregulatory cytokines". The manuscript, in general, is well written, The topic is of great interest.

Further comments:  

  • Sensor performances, such as linearity range, limit of detection (LOD), limit of quantification (LOQ), repeatability, intermediate precision, selectivity, and lifetime, should be discussed in the review. Authors should compare the performance of different sensors.   

Thank you for the suggestion. We have added a Table to compare different nanomaterials in section 5. Please see Table 4, Page 24-25. Since the information on detection range and detection limit has been mentioned across the text, we decided to construct the table to give the information on important features of each nanomaterial instead.

  • Authors described graphene, fullerenes, CNTs etc. as good candidates for biosensor applications. Authors should cite recent relevant reviews there.  Recent Advances in Sensing Applications of Graphene Assemblies and Their Composites. Adv. Funct. Mater. 2017, 27, 1702891. Doi:10.1002/adfm.201702891; and Carbon Nanomaterial Based Biosensors for Non-Invasive Detection of Cancer and Disease Biomarkers for Clinical Diagnosis. Sensors 2017, 17, 1919; DOI:10.3390/s17081919.

The suggested articles have been cited. They are reference 115 and 116 in the revised manuscript.

Reviewer 3 Report

Comments to the authors

The manuscript reviews the evolving nanomaterials-based biosensors for cytokines detection as well as the strategies and limitations associated with these biosensors. Major signal transduction methods employed in nanomaterials-based biosensors including optical, electrochemical and magnetic transduction methods were mentioned. Nanomaterials discussed are QDs, noble metal nanomaterials, metal oxide nanomaterials, carbon-based nanomaterials, and bio-nanomaterials. However the following issue should be addressed before it could be published:

This work is given as a review, but the pictures for descripting the reviewed works could not be found in the manuscript. Please attach some pictures about the action mechanism of different nanomaterials. And the pictures should be reasonable and useful, or else the current version is not a review indeed and maybe a perspective.

Author Response

Reviewer 3

The manuscript reviews the evolving nanomaterials-based biosensors for cytokines detection as well as the strategies and limitations associated with these biosensors. Major signal transduction methods employed in nanomaterials-based biosensors including optical, electrochemical and magnetic transduction methods were mentioned. Nanomaterials discussed are QDs, noble metal nanomaterials, metal oxide nanomaterials, carbon-based nanomaterials, and bio-nanomaterials. However, the following issue should be addressed before it could be published:

This work is given as a review, but the pictures for descripting the reviewed works could not be found in the manuscript. Please attach some pictures about the action mechanism of different nanomaterials. And the pictures should be reasonable and useful, or else the current version is not a review indeed and maybe a perspective.

We are thankful for your suggestion. We have added 2 figures to illustrate the concepts of biosensors from AuNPs and AgNPs described in the article. Please see Figure 2 (Page 12) and Figure 3 (Page 13) in the revised manuscript. We also added 2 more tables to summarize carbon-based cytokine biosensor discussed in the review and described the advantages and disadvantages of each nanomaterials for cytokine biosensors application. Please see Table 3 (Page 21-22) and Table 4 (Page 24-25) in the revised manuscript.

Reviewer 4 Report

This is a review of the advancement in nanotechnology-based biosenors for immunoregulatory cytokines.  A wide variety of nanomaterials are covered including quantum dots, noble metal nanomaterials, metal oxide nanomaterials, and carbon-based nanomaterials.  It is an important topic and is of interest to readers in the field of bio-nanotechnology.  However, several improvements are recommended before publication:

  1. The manuscript should review prior review articles on the related topic and indicate the differences between this paper and previous published review articles.
  2. Use figures to provide better descriptions of the examples cited in the manuscript.
  3. Add one or more Tables to summarize the different types of new nanomaterial-based biosensors and compare their specifications.  There is a Table for the metal oxide nano-sensors but no similar Tables for other nanomaterial-based sensors.   
  4. Include more detailed specifications in the Tables that compare different types of nanomaterial-based sensors such as detection limit, linear range, applicable sample matrix, analytical device needed for the analysis, etc.
  5. Provide more information about the stage of development of the examples.

Author Response

Reviewer 4

This is a review of the advancement in nanotechnology-based biosenors for immunoregulatory cytokines.  A wide variety of nanomaterials are covered including quantum dots, noble metal nanomaterials, metal oxide nanomaterials, and carbon-based nanomaterials.  It is an important topic and is of interest to readers in the field of bio-nanotechnology.  However, several improvements are recommended before publication:

  • The manuscript should review prior review articles on the related topic and indicate the differences between this paper and previous published review articles.

We thank the reviewer for the suggestion to improve this manuscript. Previous reviews on related topic have been discussed in the introduction part. Please see Page 2, line 69-70; Page 3, line 71-73 and 78-82 in the revised manuscript.

  • Use figures to provide better descriptions of the examples cited in the manuscript.

We have added 2 figures to illustrate the concepts of some biosensors from AuNPs and AgNPs described in the article. Please see Figure 2 (Page 12) and Figure 3 (Page 13) in the revised manuscript.

  • Add one or more Tables to summarize the different types of new nanomaterial-based biosensors and compare their specifications.  There is a Table for the metal oxide nano-sensors but no similar Tables for other nanomaterial-based sensors.   

We have added Table 3 to summarize carbon-based cytokine biosensors discussed in the review. Please see Page 21-22 of the revised manuscript.

  • Include more detailed specifications in the Tables that compare different types of nanomaterial-based sensors such as detection limit, linear range, applicable sample matrix, analytical device needed for the analysis, etc.

Table 4 has been added to the revised manuscript (Page 24-25) to provide information on important features of each nanomaterial. Since information on detection limit and range were mentioned throughout the manuscript, we chose to focus on the advantages and disadvantages of each nanomaterials.

  • Provide more information about the stage of development of the examples.

A paragraph to discuss the stage of development of cytokine biosensors in general has been added at the end of section 5. Please see Page 24, line 944-956 in the revised manuscript.

Round 2

Reviewer 3 Report

The author has revised the manuscript according to the reviewer's comments, and the revised version could be published in Biosensors.

Reviewer 4 Report

The authors have made all the changes requested.  The publication of the manuscript is recommended.